# Comparative Evaluation of the Clinical Severity of COVID-19 of Vaccinated and Unvaccinated Patients in Southeastern Romania in the First 6 Months of 2022, during the Omicron Wave

**DOI:** 10.3390/healthcare11152184

**Published:** 2023-08-01

**Authors:** Corina Manole, Liliana Baroiu, Aurel Nechita, Doina Carina Voinescu, Anamaria Ciubara, Mihaela Debita, Alin Laurențiu Tatu, Alexandru Bogdan Ciubara, Ioana Anca Stefanopol, Lucreția Anghel, Alexandru Nechifor, Dorel Firescu

**Affiliations:** 1Clinical Medical Department, Faculty of Medicine and Pharmacy, ‘Dunarea de Jos’ University, 800008 Galati, Romania; corina.manole@ugal.ro (C.M.); nechitaaurel@yahoo.com (A.N.); carinavoinescu@gmail.com (D.C.V.); anamburlea@yahoo.com (A.C.); dralin_tatu@yahoo.com (A.L.T.); anghel_lucretia@yahoo.com (L.A.); alexandrunechiformed@yahoo.com (A.N.); 2‘Sf. Apostol Andrei’ Clinical Emergency County Hospital, 800578 Galati, Romania; alexandru.ciubara@ugal.ro (A.B.C.); dorelfirescu@yahoo.com (D.F.); 3‘Sf. Cuv. Parascheva’ Clinical Hospital of Infectious Diseases, 800179 Galati, Romania; debita_mihaela@yahoo.com; 4‘Sf. Ioan’ Clinical Hospital for Children, 800487 Galati, Romania; ancaflorea1969@yahoo.com; 5‘Elisabeta Doamna’ Psychiatric Hospital, 800179 Galati, Romania; 6Medical Department, Faculty of Medicine and Pharmacy, ‘Dunarea de Jos’ University, 800008 Galati, Romania; 7Multidisciplinary Integrated Center of Dermatological Interface Research MIC-DIR, 800010 Galati, Romania; 8Clinical Surgical Department, ‘Dunarea de Jos’ University of Galati, 800008 Galati, Romania

**Keywords:** COVID-19, Omicron, vaccination, hospitalization, prognostic

## Abstract

(1) Background: The pandemic wave produced by SARS-CoV-2 Omicron was characterized by milder clinical forms and high contagiousness. The vaccination rate against COVID-19 in Romania was approximately 42%. (2) Objectives: Comparison of the clinical severity in vaccinated patients compared to unvaccinated ones. (3) Methods: A retrospective cohort study was conducted on a group of 699 adult patients confirmed with COVID-19 who presented in the “Sf. Cuvioasa Parascheva” Infectious Diseases Clinical Hospital of Galati, Romania, between 1 January 2022 and 30 June 2022. The study compared the need for hospitalization, reinfections, demographic and comorbidity data, clinical and paraclinical parameters from the initial evaluation, and the ratio of unfavorable developments on subgroups chosen according to the vaccination status. (4) Results and Conclusions: Our study reveals that unvaccinated patients required hospitalization in 54.68% of cases, while fully vaccinated patients had a hospitalization rate of 40.72%, which was significantly lower than that of the unvaccinated group (*p* = 0.01); patients who received a booster dose had a hospitalization rate of 27.84% (*p* < 0.01, significantly lower than unvaccinated individuals; *p* = 0.01, significantly lower than fully vaccinated individuals); and among the four patients who received four doses, none required hospitalization. From the analysis of the two subgroups of hospitalized patients, we observed a significantly higher prevalence of radiological lesions, such as pulmonary opacities in the group of unvaccinated patients and a higher average duration of hospitalization, and serum values of D-dimers and blood-sugar at admission were significantly higher in unvaccinated patients. The higher presence of these parameters, which are indicators of severe progression in clinical studies, in the group of unvaccinated patients suggests the need to include them in the initial evaluation of the unvaccinated patients with COVID-19.The cumulative share of deaths and transfers in the ICU was higher in the group of unvaccinated patients, but the difference between the groups had no statistical significance. This study draws attention to the possibility of severe clinical forms among both vaccinated and unvaccinated populations, especially in the elderly and in patients with multiple comorbidities.

## 1. Introduction

In Romania, the first SARS-CoV-2 Omicron strain was isolated on 4 December 2021 in two patients who arrived from South Africa [1]. By the end of 2021, the Romanian Ministry of Health confirmed 92 cases of infections with the Omicron variant by randomly performing genetic tests for viral identification in hospitalized patients with COVID-19, and it confirmed community transmission of this variant and anticipated that it would become the dominant variant in the next two weeks [2].

Until 3 July 2022, the National Institute of Public Health in Romania reported 14,590 SARS-CoV-2 strain sequences, of which 13,978 strains corresponded to variants of concern (VOC), and, of these, there were 1722 strains of Alpha, 11 of Beta, 23 of Gamma, 5958 of Delta, and 6264 of Omicron. Among the Omicron strains, 2845 (45%) were BA.2 and 58 (0.93%) were BA.5.

As for the number of cases, as reported by the National Institute of Public Health in Romania between 2 January 2022 and 3 July 2022, there were 1,114,131 illnesses (38.06% of the total cases reported until 3 July 2022) and 6943deaths (10.55% of the total number of deaths reported until 3 July 2022). Thus, the Omicron pandemic wave in Romania was characterized by a large number of infections per day and a lower death rate than previous pandemic waves [3].

Clinically, increased contagiousness has been described (Omicron spreads up to 3.31 times faster than the Delta variant), but with lower severity compared to Delta [4,5,6]. Most studies found that the length of the period of communicability for Omicron BA.1 (3–5 days post-symptom onset (PSO)) was shorter than for the wild-type strain (3–8 days PSO) [7].The reported Omicron range of mean incubation period (2.5–4.6 days) was shorter than that of other variants of concern (VOC) (3.3–6.5 days) [4,5,6,7,8].

In the in vitro studies on human cell cultures, the Omicron variant showed a significantly reduced neutralization of the antibodies in the serum of the patients vaccinated with two doses of the vaccine, a conserved T-cells response, and a preserved sensitivity to the antiviral drugs that target the polymerase (Remdesivir, Molnupiravir). Omicron also manifested a broad resistance to clinically-approved monoclonal antibodies (Casivirimab, Imdevimab) [9].

Clinical trials provide evidence that vaccine booster doses protect against hospitalization and death, including in COVID-19 cases with the Omicron variant [10,11,12]. Omicron infection compared to Delta was associated with greater survival [13].

Thus, we started from the hypothesis of a milder clinical course of COVID-19 cases with the Omicron variant in vaccinated patients compared to unvaccinated ones. The aim of the study was to highlight the particularities of the Omicron pandemic wave in our region. We aimed to identify clinical and paraclinical parameters from the initial evaluation of the patient that may suggest an unfavorable course of COVID-19. We also aimed to quantify the percentage of unfavorable evolutions in a population with lower vaccination coverage than the states where early studies were conducted [14,15,16,17,18,19] in order to assess the clinical severity of this disease in our region. 

We conducted a retrospective cohort study between the 1 January 2022 and the 30 June 2022 in an Infectious Diseases Department of a hospital in Southeastern Romania.

## 2. Materials and Methods

A retrospective cohort study was performed on a group of 699 adult patients, representing all confirmed adult COVID-19 cases, either through rapid antigen testing or RT-PCR, admitted to “Sf. Cuvioasa Parascheva” Infectious Diseases Clinic Hospital in Galati, Romania between the 1 January 2022 and the 30 June 2022 for the diagnosis of COVID-19.

The inclusion criteria (Figure 1):-Total group—699 patients—adults with rapid antigen test or RT-PCR for SARS-CoV-2 positive.-Group A-304 patients hospitalized for severe forms of COVID-19 or with non-severe forms but with multiple comorbidities who required in-hospital treatment during the development of acute SARS-CoV-2 infection.-Group B-395 outpatients with non-severe forms who were clinically assessed through the outpatient department, underwent minimal biological and radiological investigations, and who received free home-based therapy with Molnupiravir for 5 days along with symptomatic treatment.-Subgroup A1-129 hospitalized patients who had been vaccinated against SARS-CoV-2 with at least one dose of the vaccine.-Subgroup A2-172 hospitalized patients who were unvaccinated.-Three patients from group A were excluded as they were foreign citizens for whom vaccination status could not be documented.-Subgroup B1-250 vaccinated patients against SARS-CoV-2 with at least one dose of the vaccine.-Subgroup B2-145 patients who were unvaccinated.

**Figure 1 healthcare-11-02184-f001:**
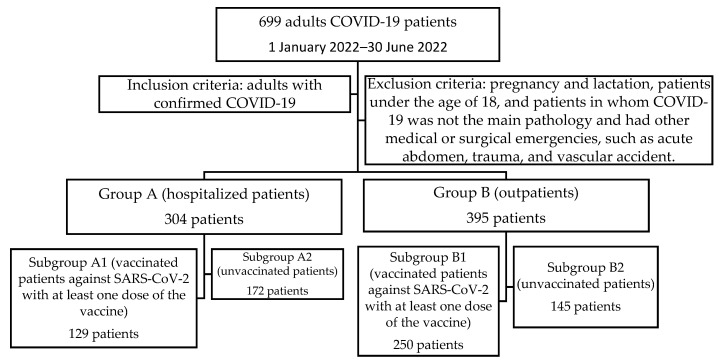
Patient groups selection.

Severe COVID-19 was defined by the presence of any of the following criteria: oxygen saturation less than 90% in room air or signs of severe respiratory distress (respiratory rate greater than 30 breaths per minute, use of accessory muscles or inability to complete full sentences) [20].

Regarding SARS-CoV-2 vaccination status, we considered the vaccinated patients to be those who had received at least one dose of the vaccine; fully vaccinated patients were those who had received a single dose of the Johnson & Johnson vaccine or two doses of Pfizer, Moderna, or AstraZeneca; boosted patients were those who had been fully vaccinated patients along with an additional dose of Pfizer or Moderna; and the vaccinated patients against the Omicron variant were the boosted patients who had also received a dose of the Pfizer bivalent vaccine.

The exclusion criteria were as follows: pregnancy and lactation; patients under the age of 18; and patients in whom COVID-19 was not the main pathology and had other medical or surgical emergencies, such as acute abdomen, trauma, or vascular accident.

The patients were monitored during their hospitalization.

The design and reporting of this study followed the Strengthening the Reporting of Observational Studies in Epidemiology (STROBE) statement [21].

Demographic, clinical, and paraclinical data were extracted from patients’ observation charts, which were obtained during the initial evaluation upon admission, and data regarding vaccination history and reinfections were extracted from the records of the Public Health Department. The data of all of the patients who came to our clinic during the study period were analyzed.

We must mention that not all of the patients had the complete set of data available, but the missing data on the presented parameters were below 2%. We specify that for certain parameters such as IL6, which was performed only in a small part of the patient cohort, this was not commented on in the results, even though it is recognized as a marker of unfavorable evolution of COVID-19 [22].

The endpoint of the study was the total number of patients who required hospitalization and the total number of patients with unfavorable evolution (considered as transfer to the intensive care unit (ICU) or death during hospitalization).

The tests used were the following: RT-PCR COVID-19-GeneXpert, Xpert x pressCoV-2 plus test, Cepheid; Antigen SARS-CoV-2-COVID-19 Antigen Rapid Test Swab Hangzhou All Test Biotech Co., Ltd., Lotus, NL, USA; Biochemistry-ABX Pentra 400, Horiba; and Hematology-ABX Pentra 80 XL, Horiba. Interstitial lung injury or alveolar opacity were assessed by a specialist radiologist on the chest X-ray, and Curb-65 score of pneumonia severity was assessed on Mdcalc [23].

Patient data extracted from the observation charts were analyzed using the IBM SPSS Statistics 26 program. Gross descriptive statistical parameters were calculated for all variables, at which this analysis was useful. The presentation is in Mean ± S.D. (standard deviation) for continuous variables and in absolute frequency (relative frequency) for categorical variables, with the test used in the inferential analysis being specified for each variable.

Comparisons were made between group A and group B and between the subgroup A1 and A2. A level of statistical significance was considered for values below 0.05 (with *p*-index calculated at both ends).

## 3. Results

Two comparative analyses were conducted. The first analysis compared group A, consisting of patients treated in the hospital (304 patients, 43.49%), with group B, consisting of patients receiving home-based treatment (395 patients, 56.50%). The second analysis compared subgroup A1, comprising vaccinated patients admitted to the hospital with at least one dose of the SARS-CoV-2 vaccine (129 patients, 42.43% of total hospitalized patients), with subgroup A2, which consisted of unvaccinated patients admitted to the hospital (175 patients, 57.56% of total hospitalized patients).

The comparative analysis between group A and B showed the following:-The median age of group A (67 years) was higher than median age of group B (55 years), and the average age in group A (62.9 years) was statistically significant and higher (*p* < 0.01) than in group B (55.36 years) (Table 1).

-The female gender was predominant in both group A and group B (Table 2).

This disease episode was a reinfection with SARS-CoV-2 in 27 of the hospitalized patients from group A (8.88%), and 50 of the cases were patients who received home-based treatment and form group B (12.65%).

In group A, 101 patients were vaccinated with BioNTech, Pfizer vaccine, of which 1 patient with one dose had zero reinfections; 58 patients with two doses presented 7 reinfections (12.06%), 42 patients with three doses presented 2 reinfections (4.76%); 18 patients vaccinated with one dose of the Johnson & Johnson vaccine experienced 1 reinfection (5.55%); 2 patients vaccinated with AstraZeneca had two doses with 1 patient experiencing reinfection (50%); 1 patient vaccinated with Moderna had two doses and no reinfections; and 7 patients vaccinated with several types of vaccine presented no reinfections.

In group B, 191 patients were vaccinated with BioNTech, Pfizer vaccine, of which 4 patients with one dose presented and 1 patient suffered reinfection (25%); 80 patients with two doses had 13 reinfections (16.25%); 105 patients with three doses, and 4 had reinfections (3.80%); 2 patients with four doses had no reinfections. Meanwhile, of the 27 patients with one dose of the Johnson & Johnson vaccine, 2 had reinfections (7.40%), and of the 4 patients who were vaccinated with AstraZeneca with 2 doses, 2 of them had reinfections (50%); and of the 9 patients vaccinated with Moderna, 3 patients who had received two doses presented 0 reinfections, whilst of the 6 patients who had received three doses, 1 presented with reinfection (16.66%). Of the 19 patients who had been vaccinated with several types of vaccine, 1 suffered reinfection (5.26%).

Thus, among the 699 patients, 320 patients were unvaccinated, 379 received at least one dose, 194 patients received complete vaccination, 176 patients received the booster dose, and 4 patients received four doses (including a bivalent Pfizer dose). Among the 320 unvaccinated patients, 54.68% required hospitalization. Among the 379 patients vaccinated with at least one dose, 34.03% required hospitalization. Among the 194 patients who received complete vaccination, 40.72% required hospitalization (*p* = 0.01, significant difference in proportion compared to those unvaccinated). Among the 176 patients who received the booster dose, 27.84% required hospitalization (*p* < 0.01, weight difference from unvaccinated, *p* = 0.01 significant difference in proportion compared to those with full vaccination). Finally, among the four patients who had received four doses, none required hospitalization.

In the following results section, we present the findings of our research on the risk estimates and the odds ratios for various cohorts in relation to vaccination status. The data were analyzed using 95% confidence intervals to assess the statistical certainty in the risk differences between vaccinated and non-vaccinated individuals within each cohort.

The overall odds ratio for being hospitalized in the vaccinated group compared to the non-vaccinated group was 0.399 (95% CI: 0.295 to 0.539), which indicates a statistically significant reduced risk of hospitalization for vaccinated individuals. Subgroup analysis revealed that among individuals not previously hospitalized, the odds ratio was 0.657 (95% CI: 0.570 to 0.758), which further supports a reduced risk of hospitalization for the vaccinated cohort. However, for individuals previously hospitalized, the odds ratio was 1.649 (95% CI: 1.396 to 1.947), indicating a statistically significant increased risk of hospitalization in the vaccinated group.

The overall odds ratio for reinfection between vaccinated and non-vaccinated individuals was 0.690 (95% CI: 0.430 to 1.106). The confidence interval includes 1, suggesting no statistical certainty in the risk difference between the two groups for reinfection. Subgroup analysis for individuals not reinfected showed an odds ratio of 0.960 (95% CI: 0.912 to 1.011), indicating no significant difference in reinfection risk between vaccinated and non-vaccinated individuals. For individuals experiencing reinfection, the odds ratio was 1.393 (95% CI: 0.914 to 2.123), which also did not reach statistical significance.

The overall odds ratio for gender between vaccinated and non-vaccinated individuals was 0.854 (95% CI: 0.632 to 1.154), and the confidence interval spans 1, suggesting no statistical certainty in the risk difference based on gender. Subgroup analysis for males and females further supported this finding, showing no statistically significant differences in the risk between the vaccinated and non-vaccinated cohorts.

The overall odds ratio for the environment of origin between vaccinated and non-vaccinated individuals was 0.371 (95% CI: 0.245 to 0.563), which indicates a statistically significant reduced risk of vaccination in certain environments. Subgroup analysis revealed that individuals from urban areas had an odds ratio of 0.853 (95% CI: 0.797 to 0.913), which demonstrates a reduced risk of vaccination, whereas individuals from rural areas had an odds ratio of 2.296 (95% CI: 1.611 to 3.273), which indicates a statistically significant increased risk of vaccination.

Regarding the share of reinfections, among the 320 unvaccinated patients, 41 (12.81%) were experiencing a reinfection; among the 194 patients with full vaccination, in 27 patients (13.91%, *p* = 0.82 compared to the unvaccinated), this episode represented a reinfection; among the 176 fully vaccinated plus booster patients, 41 (23.29%, *p* = 0.01, versus unvaccinated) had a reinfection; and among the 4 patients vaccinated with four doses, including a bivalent vaccine, none experienced a reinfection.

The second comparative analysis between subgroups A1 and A2 revealed that the mean age in subgroup A1 was 61.8 years, with no statistically significant difference with subgroup A2, where it was 64 years (Table 3).

In this section, we present the results of our study, examining the risk estimates and odds ratios for different cohorts of hospitalized patients in relation to various factors, including sex, living area, unfavorable evolution, antiviral treatments, immunosuppressive treatment, and imaging changes. The statistical certainty was assessed using 95% confidence intervals.

For both male and female cohorts, the risk estimates did not show statistically significant differences between the vaccinated and the unvaccinated individuals in terms of hospitalization. The confidence intervals encompassed the value 1, indicating no significant risk variation.

The “Environment = U” cohort showed a significantly reduced risk (OR = 0.291) for unfavorable evolution among vaccinated individuals living in this area. Conversely, the “Environment = R” cohort revealed a significantly increased risk (OR = 2.615) of unfavorable evolution for vaccinated individuals in this living area (Table 4).

-The nutrition status at admission and the average body mass index in group A1 is 27.33, without a statistically significant difference from group A2 in which it was 28.32 (Table 3).-The average Charlson score of cumulative comorbidities in group A1 (2.43) is without a statistically significant difference compared to group A2 (2.66) (Table 3).-The average Curb 65 score of pneumonia severity in group A1 (1.51) is without a statistically significant difference compared to group A2 (1.55) (Table 3).-The average number of days of hospitalization in group A1 (5.883 days) without a statistically significant difference compared to group A2 (6.76 days) (Table 3).

There were no significant differences in risk for both “Unfavorable evolution = No” and “Unfavorable evolution = Yes” cohorts, as indicated by confidence intervals spanning 1 (Table 4).We note that in the subgroup of vaccinated patients, out of the four cases transferred to the ICU, two cases were in fully vaccinated patients and two cases were in boosted vaccinated patients.

Both “Remdesivir” and “Favipiravir or Molnupiravir” cohorts did not exhibit statistically significant differences in risk between vaccinated and non-vaccinated individuals. The confidence intervals included 1 for all subgroups.

Our findings revealed no statistically significant variations in risk for both “Immunosuppressive treatment = No” and “Immunosuppressive treatment = Yes” cohorts.

For both “Alveolar pulmonary opacity” and “Accentuated pulmonary interstitial” cohorts, there were no statistically significant differences in risk between vaccinated and non-vaccinated individuals (Table 4).

-The average oxygen saturation at admission did not show a statistically significant difference between the two subgroups of hospitalized patients (Table 3).-Regarding the average values of the inflammatory markers (C-reactive protein, fibrinogen, and erythrocyte sedimentation rate), no statistically significant differences were observed between the two subgroups of hospitalized patients. Similarly, in the average value of serum procalcitonin, there was also no statistically significant difference between the groups (Table 3).

**Table 3 healthcare-11-02184-t003:** Comparative values of COVID-19 characteristics of patients hospitalized with COVID-19 and vaccinated against SARS-CoV-2 compared to those hospitalized with COVID-19 and not vaccinated against SARS-CoV-2.

	Vaccinated Patients N = 129 (A)				Not Vaccinated Patients N = 172 (B)				*p* (*t*-Test)
	Av.	SD	Min; Max	Median	Av.	SD	Min; Max	Median	
Age (years old)	61.8	16.8	18; 90	67	64	18.4	18; 96	68	0.28
Hospitalization length (days)	5.88	3.21	1; 19	5	6.75	3.60	1; 21	6	0.03
Charlson score	2.43	1.77	0; 7	3	2.66	1.78	0; 8	3	0.27
Oxygen saturation at admission (%)	94.75	3.73	60; 98	95	94.28	2.52	86; 98	95	0.19
D-dimer (ng/mL)	855.47	1264.90	130.6; 10,000	512.86	1390.02	1955.01	45; 12,226.32	798.27	0.01
CK (U/L)	181.04	324.08	14.2; 2477.2	86.8	363.8	2023.65	5.9; 25,125	97.5	0.33
Troponin (ng/mL)	36.72	228.09	1.5; 2171	4	64.01	294.6	1.5; 2947	5.8	0.45
ESR (mm/1h)	31.01	21.7	4; 90	24	36.12	23.71	2; 110	30	0.06
C-reactive protein (mg/L)	33.71	51.69	0.18; 378.09	11.85	39.94	51.17	0.18; 301.4	24.32	0.32
Leukocytes (N/µL)	7003.88	5420.67	1220; 47,000	6100	6449.71	6890.95	1020; 81,600	5400	0.45
Platelets (N/µL)	226,054.26	85,583.69	24,000; 683,000	213,000	205,575.58	82,571.96	36,000; 511,000	193,000	0.04
Hb (g/dL)	13.45	1.61	8.3; 17.2	13.6	13.1	1.74	7.5; 19.9	13.15	0.08
ALT (U/L)	30.61	24.87	7.2; 162.9	22.4	47.76	210.8	6.1; 2736.5	24.05	0.36
AST (U/L)	30.68	17.41	12.8; 130.8	24.6	52.4	132.17	10.5; 1578.4	29.95	0.07
Total bilirubin (mg/dL)	0.59	0.36	0.13; 2.74	0.52	0.74	1.32	0.15; 13.36	0.5	0.20
Direct bilirubin (mg/dL)	0.19	0.12	0.05; 0.74	0.16	0.34	1.16	0.04; 11.8	0.17	0.16
Prothrombin concentration (%)	95.72	18.61	28.5; 130	97.9	92.53	17.83	20.4; 129.5	96.2	0.15
Serum urea (mg/dL)	34.94	17.15	11.5; 116.4	30.3	36.75	20.22	8.3; 146.3	30.7	0.42
Serum creatinine (mg/dL)	1.03	0.5	0.52; 4.87	0.93	1.04	0.61	0.51; 7.25	0.9	0.93
Serum glucose (mg/dL)	106.26	27.69	26.2; 219.3	97.8	122.15	53.08	53.2; 357.7	105.65	0.01
BMI	27.328	6.302	17.04; 57.1	26.3					
Number of days required for oxygen therapy	0.52	1.89	0; 10	0	0.96	2.89	0; 19	0	0.13
CURB 65 score	1.51	0.64	0; 3	2	1.55	0.65	0; 3	2	0.64

Abbreviations and normal values: Av, Average; SD, Standard Deviation; Max, Maximum Value; Min, Minimum value; T-*t*, Student Test (*t*) for differences between means; Normal values: D-dimer < 500 ng/mL; CK, Creatine kinase 0–171 U/L; troponin 0.01–0.1 ng/mL; ESR, Erythrocytes sedimentation rate 0–15 mm/1 h; C-reactive protein < 10 mg/L; leukocytes 4000–10,000/µL; Hg, hemoglobin 13–17 g/dL; Platelets 150–450 × 10^6^/µL; ALT, alanil-aminotransferase 0–45 U/L; AST, aspartate-aminotransferase 0/35 U/L; Total bilirubin 0.1–1.2 mg/dL; Direct bilirubin 0–0.2 mg/dL; Prothrombin concentration 70–100%;Serum urea 18–55 mg/dL; Serum creatinine 0.8–1.3 mg/dL; Serum glucose 74–108 mg/dL; BMI, body mass index 18.5–24.9 kg/m^2^.

**Table 4 healthcare-11-02184-t004:** Characteristics of categorial variables of the patients hospitalized with COVID-19 and vaccinated against SARS-CoV-2 compared to those hospitalized with COVID-19 and not vaccinated against SARS-CoV-2.

Risk Estimate and Odds Ratios for Hospitalized Patients			
Sex and Vaccination Status	Value	95% Confidence Interval	
		Lower	Upper
Odds Ratio for Vaccinated(No/Yes)	0.895	0.573	1.399
For cohort Sex = M	0.939	0.730	1.208
For cohort Sex = F	1.049	0.864	1.275
Living Area and vaccination status			
Odds Ratio for Vaccinated (No/Yes)	0.291	0.161	0.525
For cohort Environment = U	0.761	0.675	0.858
For cohort Environment = R	2.615	1.609	4.251
Unfavorable evolution (death or transfer of intensive care) andvaccination status			
Odds Ratio for Vaccinated (No/Yes)	0.805	0.264	2.458
For cohort Unfavorable evolution (death or transfer of intensive care) = No	0.991	0.946	1.038
For cohort Unfavorable evolution (death or transfer of intensive care) = Yes	1.231	0.422	3.591
Antiviral treatment and vaccination status			
Remdesivir			
Odds Ratio for Vaccinated (No/Yes)	1.184	0.620	2.262
For cohort Remdesivir = No	1.023	0.937	1.117
For cohort Remdesivir = Yes	0.864	0.494	1.512
Favipiravir or Molnupiravir			
Odds Ratio for Vaccinated (No/Yes)	1.038	0.642	1.677
For cohort Favipiravir or Molnupiravir = No	1.026	0.735	1.432
For cohort Favipiravir or Molnupiravir = Yes	0.989	0.854	1.145
Immunosuppressive treatment and vaccination status			
Odds Ratio for Vaccinated (No/Yes)	1.212	0.765	1.918
For cohort Immunosuppressive treatment = No	1.072	0.905	1.270
For cohort Immunosuppressive treatment = Yes	0.885	0.662	1.184
Imaging changes and vaccination status			
Alveolar pulmonary opacity			
Odds Ratio for Vaccinated (No/Yes)	0.823	0.418	1.622
For cohort Alveolar pulmonary opacity = No	0.976	0.899	1.060
For cohort Alveolar pulmonary opacity = Yes	1.185	0.653	2.151
Accentuated pulmonary interstitialand vaccination status			
Odds Ratio for Vaccinated (No/Yes)	1.192	0.745	1.908
For cohort Accentuated pulmonary interstitial = No	1.124	0.821	1.537
For cohort Accentuated pulmonary interstitial = Yes	0.942	0.805	1.103

-The average values of D-dimers were statistically significantly higher in group A2 (*p* = 0.01) (Table 3).-The changes found at admission in the complete blood count (CBC) tests of the hospitalized patients with COVID-19 (total leukocyte count and serum hemoglobin) did not show statistically significant differences between the A1 and A2 subgroups of patients (Table 3).-Comparing the mean values obtained in the analyzes exploring liver function at admission (ALT, AST, total and direct bilirubin, serum prothrombin concentration)did not show significant statistical differences between the two groups of hospitalized patients (Table 3).-The average values of serum urea and creatinine at admission were within the ranges of normal values of the analyses, without significant statistical differences between the two groups of hospitalized patients (Table 3).-Regarding the pancreatic damage from COVID-19, our study notes the average blood glucose values without significant statistical differences between the groups, with the values being above the maximum normal value in group A1 and slightly above the maximum normal value in group A2 (Table 3).-In the average values of creatine-kinase (CK) and troponin, there were no statistically significant differences between the two groups (Table 3).

## 4. Discussion

The aim of this study was to quantify the impact of vaccination on the clinical severity of COVID-19 during the pandemic wave driven by the Omicron variant in an Eastern European population with a lower complete vaccination rate than the Western European states, where the first studies were published. In Romania, the complete vaccination rate in June 2023 was 42% [24], and as of 1 June 2023, the National Center for Surveillance and Control of Communicable Diseases in Romania reported that 8,142,480 individuals had received their first dose of the vaccine, 8,131,090 were completely vaccinated, and 2,668,607 had received a booster dose, including 15,340 individuals who had received a dose of Pfizer Omicron [25].

Regarding the need for hospitalization, our study observed that unvaccinated patients required hospitalization in more than half of the cases (54.68%); patients with complete vaccination required hospitalization in a proportion of 40.72%, which was significantly lower than the unvaccinated group (*p* = 0.01); boosted patients required hospitalization in a proportion of 27.84% (*p* < 0.01, significantly lower than the unvaccinated group; *p* = 0.01, significantly lower than those with complete vaccination), which was significantly lower than unvaccinated patients and those with complete vaccination; and none of the four patients who had received four doses required hospitalization. These observations can be considered evidence of a lower clinical severity of COVID-19 associated with the Omicron variant in terms of hospitalization needs, particularly in patients with complete vaccination, and this was even more significant in patients with complete vaccination plus a booster shot. It is worth noting that none of the patients who received four doses required hospitalization, but the small number of patients does not allow for a conclusive statement regarding this issue.

The results of our study demonstrate significant variations in the risk of hospitalization and environment of origin based on vaccination status. Vaccinated individuals had a reduced risk of hospitalization, especially among those not previously hospitalized. However, a higher risk of hospitalization was observed for vaccinated individuals with a history of hospitalization, which could be attributed to differences in baseline health conditions or other factors not accounted for in this analysis.

In terms of reinfection, our findings do not show a statistically significant difference in the risk between vaccinated and non-vaccinated individuals, both for those who were reinfected and those who were not, and this suggests that vaccination may not have a significant impact on the risk of reinfection in the studied population.

Regarding gender, our study does not reveal any statistically significant differences in the risk of vaccination between males and females, which indicates that the effect of vaccination is not influenced by gender in this cohort.

The most notable finding pertains to the association between vaccination status and environment of origin. Vaccinated individuals from urban areas had a reduced risk of vaccination, whereas those from rural areas had a significantly increased risk. This suggests that the effectiveness of vaccination may vary based on the local environmental factors and healthcare practices in different regions.

Overall, these results contribute valuable insights into the impact of vaccination across different cohorts, and they underscore the importance of considering contextual factors when evaluating vaccination outcomes. Nevertheless, further research and validation are necessary in order to corroborate these findings and to provide a comprehensive assessment of vaccination outcomes in diverse populations.

This study also highlights the importance of providing antiviral home-based treatment for non-severe cases, which can have an impact on patient prognosis, and which can also relieve the burden on infectious disease hospitals caused by COVID-19 pathology.

From the analysis of the outpatient group treated for non-severe COVID-19 cases, we observed a younger age and a significantly higher vaccination rate (63.29%) compared to the group of hospitalized patients with severe COVID-19 or severe comorbidities requiring hospital monitoring during the acute phase of the disease, highlighting the importance of careful initial evaluation for admission decisions, especially in elderly unvaccinated patients.

A study from South Africa compared the severity of clinical disease of patients with Omicron versus strains of the previous three pandemic waves. The study noted severe forms of COVID-19: 33.6% of patients hospitalized with Omicron, 52.3% of those hospitalized in the first wave of the pandemic, 63.4% of those hospitalized in the second wave, and 63% of those hospitalized with Delta [26].

A study conducted on a cohort of Scottish patients affected by the infection with the Omicron variant from November to December in 2021 suggested that Omicron is associated with a two-thirds reduction in the risk of hospitalization compared to the Delta variant, which indicates that a booster dose compared to the second dose of the vaccine provides substantial additional protection against the risk of symptomatic COVID-19 for Omicron [27].

A clinical study conducted in Denmark on a cohort of COVID-19 patients affected by the Omicron variant during the period of April to June 2022 observed the differences in clinical severity caused by different variants of Omicron. The study’s observations were as follows: previous infection with Omicron provided high protection against infections with the BA.5 and BA.2 strains in triple-vaccinated individuals; vaccine protection against BA.5 infection was similar or slightly weaker than protection against BA.2 infection; and BA.5 infections were associated with a higher risk of hospitalization compared to BA.2 infections [28].

A study conducted in Hong Kong on 5310 subjects between January and July in 2022 observed that three and four doses of BNT162b2 or CoronaVac were effective in preventing Omicron infection, starting from 7 days after vaccination [29].

From the analysis of the two subgroups of hospitalized patients, we observe a significantly higher proportion of radiological lesions of pulmonary opacities in the group of unvaccinated patients as well as higher average values of hospitalization duration, serum levels of D-dimers, and blood glucose at admission, all of which are significantly higher in unvaccinated patients. All these parameters are considered, according to the results of various clinical studies, to be markers of a more severe disease progression [30,31,32]. Therefore, the higher proportion of these parameters indicating severe disease progression in the group of unvaccinated patients suggests the necessity of including chest X-ray and serum D-dimer and blood glucose measurements in the initial evaluation of unvaccinated patients with COVID-19.

The results of our study suggest that the risk of hospitalization and unfavorable evolution does not significantly differ between vaccinated and non-vaccinated individuals based on sex, imaging changes, antiviral treatments, and immunosuppressive treatment cohorts. These findings align with previous research that has shown the overall effectiveness of vaccination in reducing severe outcomes and hospitalization rates.

However, an interesting observation was made concerning living area and vaccination status. Vaccinated individuals residing in urban areas demonstrated a significantly reduced risk of unfavorable evolution, while those in rural areas exhibited a notably increased risk. This implies that contextual factors, such as environmental conditions and healthcare resources, may influence vaccination outcomes in different regions.

Overall, our study emphasizes the importance of considering contextual factors when assessing the impact of vaccination on hospitalized patients. Nevertheless, further research is warranted to validate and expand upon these findings. A comprehensive understanding of these factors will aid in refining vaccination strategies and optimizing healthcare interventions to reduce the burden of severe outcomes in hospitalized patients, as other studies have concluded [33,34].

The cumulative proportion of deaths and transfers to the ICU was higher in the group of unvaccinated patients, although the difference between groups was not statistically significant. The study draws attention to the possibility of severe clinical forms occurring both in vaccinated and unvaccinated populations, especially in the elderly and patients with multiple comorbidities.

These results motivate us to promote vaccination and support the development of vaccines adapted to circulating strains, and they also draw attention to the possibility of a more severe progression of COVID-19 in unvaccinated patients and the occurrence of clinically unfavorable forms in both vaccinated and unvaccinated populations, especially among the elderly with multiple comorbidities.

A study from India on 290 patients who experienced Omicron from 24 November 2021 to 4 January 2022 noted the presence of one death (0.3%) and severe forms of the disease (0.7%) [35]. A study from South Africa observed that the mortality rate in hospitalized patients was 10.7% for Omicron, 21.5% for Wuhan, 28.8% for Wave 2, and 26.4% for Delta [26]. The proportion of deaths and transfers to the ICU observed in our study was 2.002% out of a total of 699 patients.

Although the Director-General of the WHO declared on 5 May 2023 that COVID-19 is now a stable and ongoing health issue that no longer constitutes a public health emergency of international concern [36], the possibility of severe clinical forms, especially in elderly patients with multiple comorbidities, justifies the continued interest in improving vaccination strategies and refining diagnostic and treatment methods for these patients.

Regarding the proportion of reinfections, the paradoxical observations of the study, with higher rates as the number of doses administered increases, cannot be interpreted due to the lack of investigation into the data related to the time that has elapsed since the last vaccine dose to the reinfection of the patients, the viral variants that caused the initial episode of COVID-19, and the adherence of the patients to protective measures. Clinical studies note that Omicron has a higher risk of reinfection than Delta [37,38]. The severity of the infections with Omicron could also be lower due to the fact that a large number of cases are reinfections, since reinfections are known to be less severe than primary infections [39].

Regarding the occurrence of long COVID after Omicron infection, our study did not provide data as the monitoring period was limited to the hospitalization period, but a study from the United Kingdom, based on self-reported symptoms, assessed the proportion of patients with long COVID in the wave of infections with the Omicron strain (defined by having new symptoms or symptoms that have not improved or disappeared 4 weeks or more after the diagnosis of COVID-19, without another previous COVID-19 episode). The observation of this study was that among Omicron cases, 4.5% of patients had long-lasting COVID compared to Delta cases, in which 10.8% had long-lasting COVID, according to other studies [40,41,42].

The main limitation of our study is that it could not perform the identification of each strain of SARS-CoV-2 and was based on the predominant circulation of the Omicron variant during the clinical trial periods [43,44]. Other limitations of the study were the monitoring of patients only during the hospitalization period; the lack of data related to the time period between the last dose of vaccine and reinfection; and the lack of data on the identification of the strain of SARS-CoV-2 that caused the initial episode of COVID-19 in the case of patients who experienced reinfection.

Another limitation of the study was that the calculation of unfavorable outcomes was only related to patients who presented to the emergency department of our hospital and excluded patients in whom COVID-19 was not the main pathology and had other medical or surgical emergencies, such as acute abdomen, trauma, or vascular accident, who were hospitalized in the multidisciplinary emergency hospital, without taking into account the evolution of all infected patients in the region during the study period, and, as a result, the effectiveness of vaccination could not be accurately calculated. We must mention that in Romania, voluntary testing at home is possible without the obligation to report the positive or negative result, and epidemiological investigations were not conducted for positive cases, which prevented the identification of asymptomatic or mildly symptomatic cases. Testing is also conducted in pharmacies, family medical practices, and other specialty hospitals (with reporting of positive cases), with mild cases being treated symptomatically without being referred to the Infectious Diseases Department. As a result, there were positive cases during the study period that could not be quantified for the accurate calculation of vaccine effectiveness and reinfection rates. For the calculation of reinfection rates, data from the Public Health Department were used, including all reported positive cases. However, this approach does not account for asymptomatic or mildly symptomatic cases that were not tested or those who tested at home without reporting their results.

## 5. Conclusions

Complete vaccination remains an effective strategy in mitigating the impact of the Omicron variant. Administering a booster dose and using a bivalent vaccine have shown an impact on clinical severity in terms of hospitalization requirement and unfavorable prognosis (death or the need for an ICU).

Large-scale clinical studies are needed to evaluate the efficacy of bivalent vaccines against the Omicron variant in terms of infection rate, clinical severity, and prognosis, and epidemiological studies are also required in order to determine the optimal timing for a booster dose based on antibody titers and circulating strains.

Although the majority of clinical studies demonstrate the reduced severity of the Omicron variant of SARS-CoV-2 and describe milder forms of COVID-19 caused by this strain compared to previous variants, our study draws attention to the possibility of unfavorable outcomes in both vaccinated and unvaccinated patients, particularly among the elderly and those with multiple comorbidities.

Our study identifies several essential parameters in the initial evaluation of unvaccinated patients with COVID-19—namely, D-dimer levels and blood glucose, which can suggest a potentially unfavorable progression.

The results of the study highlight the need for active global vaccination policies against COVID-19 as well as continuing research for the modulation and the improvement of diagnostics, monitoring, and tailored treatment for each pandemic wave. 

## Figures and Tables

**Table 1 healthcare-11-02184-t001:** Comparative values of age of the patients hospitalized with COVID-19 and of the outpatients with COVID-19.

	Vaccinated Patients N = 304 (A)				Unvaccinated Patients N = 395 (B)				*p* (*t*-Test)
	Av.	SD	Min–Max	Median	Av.	SD	Min–Max	Median	
Age (years old)	62.9	17.9	18–96	67	55.4	15	18–93	55	0.28

Abbreviations: Av, Average; SD, Standard Deviation; Max, Maximum Value; Min, Minimum value; T-*t*, Student Test (*t*) for differences between means.

**Table 2 healthcare-11-02184-t002:** Characteristics of categorial variables of the patients hospitalized with COVID-19 compared to outpatients with COVID-19.

Risk Estimate and Odds Ratio			
	Value	95% Confidence Interval	
Vaccination status and hospitalization		Lower	Upper
Odds Ratio for Vaccinated (No/Yes)	0.399	0.295	0.539
For cohort hospitalized = No	0.657	0.570	0.758
For cohort hospitalized = Yes	1.649	1.396	1.947
Vaccination status and reinfection			
Odds Ratio for Vaccinated (No/Yes)	0.690	0.430	1.106
For cohort Reinfected = No	0.960	0.912	1.011
For cohort Reinfected = Yes	1.393	0.914	2.123
Vaccination status and gender			
Odds Ratio for Vaccinated (No/Yes)	0.854	0.632	1.154
For cohort Sex = M	0.908	0.756	1.092
For cohort Sex = F	1.063	0.946	1.195
Vaccination status and environment of origin			
Odds Ratio for Vaccinated (No/Yes)	0.371	0.245	0.563
For cohort Environment of origin = U	0.853	0.797	0.913
For cohort Environment of origin = R	2.296	1.611	3.273

## Data Availability

The data that support the findings of this study are available from the corresponding author.

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
