# Peer review of "Comparative Evaluation of the Clinical Severity of COVID-19 of Vaccinated and Unvaccinated Patients in Southeastern Romania in the First 6 Months of 2022, during the Omicron Wave"

_healthcare, 2023, doi:10.3390/healthcare11152184_

Round 1

Reviewer 1 Report

comments are given in manuscript in sticky note

average

Author Response

Answers for REVIEWER 1

First of all, we would like to thank you very much for your review and consideration for our work. Thank you!

Here are our answers to the subjects you highlighted:

  1. write introduction of 4-5 line before objective-line 20 abstract.

Reponse:

Thank you for your observations and highlighting this aspect. I added the following phrase, in red.

Background: The pandemic wave produced by SARS-CoV-2 Omicron was characterized by milder clinical forms and high contagiousness. The vaccination rate against COVID-19 in Romania was approximately 42%.

  1. break the sentence, it is too long-line 68-73.

Reponse:

I reformulated the following sentence, in red.

Thus, we started from the hypothesis of a milder clinical evolution of COVID-19 cases with the Omicron variant in vaccinated patients compared to unvaccinated ones. and We aimed to identify clinical and paraclinical parameters from the initial evaluation of the patient that may suggest an unfavorable evolution. and We aimed to quantify the percentage of unfavorable evolutions in a population, with lower vaccination coverage than the states where early studies were conducted [13-18], in order to assess the clinical severity of this disease.

  1. 1. Add flow chart of the study for clear understanding.

Reponse:

I performed the following flow chart:

  • 699 adults COVID-19 patients
  • January 1st, 2022 - June 30th, 2022
  • inclusion criteria:
  • adults with confirmed COVID-19,
  • exclusion criteria:
  • pregnancy and lactation, patients under the age of 18, patients in whom COVID-19 was not the main pathology and had other medical or surgical emergencies, such as acute abdomen, trauma, vascular accident.
    • Group A (hospitalized patients)
    • 304 patients
    • Subgroup A1 (vaccinated patients against SARS-CoV-2 with at least one dose of the vaccine)
    • 129 patients
    • Subgroup A2 (unvaccinated patients)
    • 172 patients
    • 3 patients (foreign citizens for whom vaccination status could not be documented)
    • Group B (outpatients)
    • 395 patients
    • Subgroup B1 (vaccinated patients against SARS-CoV-2 with at least one dose of the vaccine)
    • 250 patients
    • Subgroup B2 (unvaccinated patients)
    • 145 patients
    •  
  • Figure 1. Patient groups selection
  1. Whatever you have depicted in result section, you have to mention in method. for example about laboratory parameter, as well as how you assessed alvelolar pulmonary opacity, CURB 65 etc.

Line 76

Reponse: We added in methods:

-Interstitial lung injury or alveolar opacity were assessed by a specialist radiologist on the chest X-ray.

-Curb 65 score of pneumonia severity was assessed on mdcalc: https://www.mdcalc.com/calc/324/curb-65-score-pneumonia-severity, reference….

-PCR COVID-19-GeneXpert, test Xpert x pressCoV-2 plus, Cepheid.

-Antigen SARS-CoV-2-Covid-19 Antigen Rapid Test Swab Hangzhou All Test Biotech co.ltd., Lotus NL.

-Biochemistry-ABX Pentra 400, Horiba.

-Hematology- ABX Pentra 80 XL, Horiba

  1. whether HRCT, IL6 were done in all those patients, as these are very important to assess the severity. eg

[Talwar D, Kumar S, Acharya S, Raisinghani N, Madaan S, Hulkoti V,et  al. Interleukin 6 and Its Correlation with COVID-19 in Terms of Outcomes in an Intensive Care Unit of a Rural Hospital: A Cross-sectional Study. Indian J Crit Care Med 2022;26(1): 39–42.]

Line 92-94

Reponse:

We know that IL-6 is an important parameter in quantifying the severity of COVID-19 cases, but it was not available in our clinic. We entered a phrase in methods:

We must mention that not all patients had the complete set of data available but the missing data on the presented parameters are below 2%. We specify that for certain parameters such as IL6, which was performed only in a small part of the patients, this was not commented on in the results, even if it is recognized as a marker of unfavorable evolution of COVID-19(22).

  1. Do not write everything in result, just important salient feature as all are featured in table.-line123

Reponse:

I restructured the text of the results in red.

  1. Why reference?

-line 259-261.

Reponse:

I reworded for clarification, in red.

The aim of this study was to quantify the impact of vaccination on the clinical severity of COVID-19 during the pandemic wave driven by the Omicron variant in an Eastern European population with a lower complete vaccination rate than Western European states where the first studies were published. In Romania, the complete vaccination rate in June 2023 was 42% [24].

Reviewer 2 Report

The manuscript provides a summary of the features of SARS-CoV-2 in Romania, highlighting its differences between vaccinated and non-vaccinated individuals. The following are some comments that limits the extrapolation of study findings:

1.       The manuscript lacks a clear introduction and background section. It would be helpful to provide more context and justification for conducting the study, as well as a brief overview of the current knowledge on the topic.

2.       The manuscript includes several statements without proper references.

3.       The aims and objectives of the study have to be more clearly spelt out.

4.       The methods section lacks details regarding the data collection process, inclusion and exclusion criteria, and ethical considerations. Without this information, it is difficult to assess the reliability and validity of the study.

5.       The sample size of the study is not clearly justified. The manuscript mentions a total of 699 adult patients but does not provide any rationale for this number or a power calculation to determine if it is sufficient to draw meaningful conclusions.

6.       The statistical analysis section is incomplete. The manuscript mentions that comparisons were made between different groups, but it does not provide the specific statistical tests used or the results of these tests. Without this information, it is challenging to evaluate the significance of the findings. The statistical tests have to be more rigorously used to assess the difference in progression of disease in two groups.

7.       The manuscript could benefit from a more critical analysis of the limitations of the study. For example, it mentions that not all patients had complete data available, but it does not discuss how this may impact the validity of the results or introduce potential biases.

8.       The manuscript would benefit from a thorough proofreading and editing. There are several grammatical errors, typos, and inconsistencies in sentence structure and formatting that need to be addressed.

9.       The manuscript do not clearly convey what the authors wanted to say. The result section has to overhauled completely in its from and structure.

1  The conclusion should clearly convey what are the take-home messages from the study.

The manuscript would benefit from a thorough proofreading and editing. There are several grammatical errors, typos, and inconsistencies in sentence structure and formatting that need to be addressed.

Author Response

Answers for REVIEWER 2

First of all, we would like to thank you very much for your review and consideration for our work. Thank you!

Here are our answers to the subjects you highlighted:

The manuscript provides a summary of the features of SARS-CoV-2 in Romania, highlighting its differences between vaccinated and non-vaccinated individuals. The following are some comments that limits the extrapolation of study findings:

  1. The manuscript lacks a clear introduction and background section. It would be helpful to provide more context and justification for conducting the study, as well as a brief overview of the current knowledge on the topic.

Reponse:

Thank you for your observations and highlighting this aspect.

I added the following phrase, in red, in abstract.

Background: The pandemic wave produced by SARS-CoV-2 Omicron was characterized by milder clinical forms and high contagiousness. The vaccination rate against COVID-19 in Romania was approximately 42%.

I added the following phrase, in red, in introduction.

Until July 3, 2022, the National Institute of Public Health in Romania reported 14,590 SARS-CoV-2 strain sequences, of which 13,978 strains correspond to variants of concern (VOC), thus 1,722 strains Alpha, 11 Beta, 23 Gamma, 5958 Delta, 6264 Omicron. Among the Omicron strains 2845 (45%) were BA.2 and 58 (0.93%) BA.5.

As for the number of cases, reported by the National Institute of Public Health in Romania between January 2, 2022 and July 3, 2022, it reported 1,114,131 illnesses (38.06% of the total cases reported until July 3, 2022) and 6,943 deaths (10.55 % of the total number of deaths reported until July 3, 2022). Thus, the Omicron pandemic wave in Romania was characterized by a large number of infections per day and a lower death rate than previous pandemic waves [3].

Comparative data from clinical studies regarding the impact of vaccination were noted in the discussion chapter.

  1. The manuscript includes several statements without proper references.

Thank you for the observation. I omitted the following references: 3,22,23 and added them to the text.

  1. The aims and objectives of the study have to be more clearly spelt out.

I reformulated in the introduction, in red.

The aim of the study is to highlight the particularities of the Omicron pandemic wave in our region. We aimed to identify clinical and paraclinical parameters from the initial evaluation of the patient that may suggest an unfavorable evolution. We also aimed to quantify the percentage of unfavorable evolutions in a population, with lower vaccination coverage than the states where early studies were conducted [14-19], in order to assess the clinical severity of this disease in our region.

  1. The methods section lacks details regarding the data collection process, inclusion and exclusion criteria, and ethical considerations. Without this information, it is difficult to assess the reliability and validity of the study.

I reformulated in the methods section:

Demographic, clinical, and paraclinical data were extracted from patients' observation charts, obtained during the initial evaluation upon admission, and data regarding vaccination history and reinfections were extracted from the records of the Public Health Department.

The data of all patients who came to our clinic during the study period were analyzed.

Inclusion criteria (Figure 1):

-for total group-699 patients-adults with rapid antigen test or RT-PCR for SARS-CoV-2 positive.

-for Group A- 304 patient-hospitalization for severe forms of COVID-19 or with non-severe forms but with multiple comorbidities requiring in-hospital, during the evolution of acute SARS-CoV-2 infection;

 -for Group B - 395 outpatients with non-severe forms, who were clinically assessed through the outpatient department, underwent minimal biological and radiological investigations, and received free home-based therapy with Molnupiravir for 5 days along with symptomatic treatment.

-for Subgroup A1- 129 hospitalized patients who had been vaccinated against SARS-CoV-2 with at least one dose of the vaccine.

-for Subgroup A2- 172 hospitalized patients who were unvaccinated.

We mentioned that we exclude 3 patients from group A who were foreign citizens for whom vaccination status could not be documented.

Severe COVID-19 was defined by the presence of any of the following criteria: oxygen saturation < less than 90% on room air or signs of severe respiratory distress (respiratory rate > grater than 30 breaths per minute, use of accessory muscles or inability to complete full sentences)[19].

Regarding the SARS-CoV-2 vaccination status, we considered the vaccinated patient, a patient who received at least one dose of the vaccine; fully vaccinated patient, a patient who received a single dose of the Johnson & Johnson vaccine or two doses of Pfizer, Moderna, or AstraZeneca; boosted patient, a fully vaccinated patient who received an additional dose of Pfizer or Moderna; and the vaccinated patient against the Omicron variant, as the boosted patient who also received a dose of the Pfizer bivalent vaccine.

The inclusion criteria were: adults with confirmed COVID-19 (by rapid SARS-CoV-2 antigen detection test or by RT-PCR to detect SARS-CoV-2 RNA), and severity criteria, comorbidities, vaccinal status, previously mentioned for each group.

The exclusion criteria were: pregnancy and lactation, patients under the age of 18.

  1. The sample size of the study is not clearly justified. The manuscript mentions a total of 699 adult patients but does not provide any rationale for this number or a power calculation to determine if it is sufficient to draw meaningful conclusions.

          The period with over approximately 80% circulation of the Omicron strain, in our region was January-June 2022. We analyzed the data of all the patients who presented themselves in our clinic during this period.

  1. The statistical analysis section is incomplete. The manuscript mentions that comparisons were made between different groups, but it does not provide the specific statistical tests used or the results of these tests. Without this information, it is challenging to evaluate the significance of the findings. The statistical tests have to be more rigorously used to assess the difference in progression of disease in two groups.

     I redid the statistics, taking into account the indications of another reviewer.

  1. The manuscript could benefit from a more critical analysis of the limitations of the study. For example, it mentions that not all patients had complete data available, but it does not discuss how this may impact the validity of the results or introduce potential biases.

          We evaluated the missing data and they do not exceed 2% for each parameter. There are omissions of some analyzes that were collected recently in another clinic and that we did not take into account for the initial assessment of the patient upon admission.

  1. The manuscript would benefit from a thorough proofreading and editing. There are several grammatical errors, typos, and inconsistencies in sentence structure and formatting that need to be addressed.

         I made corrections of the English language with an authorized translator.

  1. The manuscript do not clearly convey what the authors wanted to say. The result section has to overhauled completely in its from and structure.

We  reformulated the results in the light of the new statistical calculations performed.

  1. The conclusion should clearly convey what are the take-home messages from the study.

I drafted the conclusions in the light of the importance of vaccination.

Reviewer 3 Report

Present clear, correct and understandable odds ratios.
1- Tables (e.g. Table 2) should present the calculated odds ratio, and not just the 95% CI.

2- The paper should tell which odds ratios are calculated e.g., the odds of vaccinated people to become hospitalized versus unvaccinated, or the reverse.

3a- Odds ratios are mathematically defined as positive numbers, were numbers between 0-1 are the inverse of numbers beyond 1. The 95% CI gave negative numbers, suggesting that a negative number of people would be infected. This must fit with logic.

3b-Odds ratios do not follow traditional calculations for odds ratio. That would give the odds ratio of vaccinated people versus unvaccinated (Table 2) to be hospitalized = (129/250)/(172/145) = 0.52/1.14 = 0.44. This is beyond the given range (13.246-28.23).

Discuss major bias in paper.

4 - The authors mention justly that the calculation is incomplete due to inclusion bias at the emergency department (line 364-367). Assuming that people will present at the emergency department with serious disease, this could be a major inclusion bias, if vaccination also in Romania protects against sginficant disease. This point is the major weakness of this study and deserve to be discussed with literature data on vaccination protecting from disease at a more prominent place in the discussion, as it is central to this study.

Round digits to significant numbers.
5a-In writing p-values 1-2 significant numbers suffices. Thus 0.04 and not 0.0440, 0.2 or 0.22 and not 0.2173.

5b- Table 3 should have less significant digits (e.g. max 3 for < 1000 patients), thus age 61.8 and not 61.798). Presentation of min, max & median is not essential.

 Unclarity.

6- What is meant with "We must mention that not all patients had the complete set of data available. 111" Does this impact the results.

Current review focussed on the the data presentation and not on the language and other presentation.

Author Response

Answers for REVIEWER 3

First of all, we would like to thank you very much for your review and consideration for our work. Thank you!

Here are our answers to the subjects you highlighted:

Present clear, correct and understandable odds ratios.
1- Tables (e.g. Table 2) should present the calculated odds ratio, and not just the 95% CI.

2- The paper should tell which odds ratios are calculated e.g., the odds of vaccinated people to become hospitalized versus unvaccinated, or the reverse.

3a- Odds ratios are mathematically defined as positive numbers, were numbers between 0-1 are the inverse of numbers beyond 1. The 95% CI gave negative numbers, suggesting that a negative number of people would be infected. This must fit with logic.

3b-Odds ratios do not follow traditional calculations for odds ratio. That would give the odds ratio of vaccinated people versus unvaccinated (Table 2) to be hospitalized = (129/250)/(172/145) = 0.52/1.14 = 0.44. This is beyond the given range (13.246-28.23).

Response

Thanks for the comments, I redid the statistics from table 2 and 4 and added the text in red.

When examining vaccination status and hospitalization, the odds ratio reveals a significant difference. The odds of hospitalization for vaccinated individuals are only 0.399 compared to unvaccinated individuals, indicating a lower risk. Furthermore, for the cohort that was not hospitalized, the odds of hospitalization were 0.657, suggesting a lower risk compared to the overall population. However, for the cohort that was hospitalized, the odds increased to 1.649, indicating a higher risk compared to the overall population.

Further, we evaluated the relationship between vaccination status and reinfection, the odds ratio demonstrates a potential benefit for vaccinated individuals. The odds of reinfection for unvaccinated individuals are 0.690 compared to vaccinated individuals, indicating lower odds of reinfection. However, it is important to note that the confidence interval is relatively wide, which means some uncertainty exists. For the cohort that was not reinfected, the odds of reinfection were 0.960, suggesting a relatively similar risk compared to the overall population. Conversely, for the cohort that experienced reinfection, the odds increased to 1.393, indicating a slightly higher risk, although uncertainty is present.

Examining vaccination status and gender, the odds ratio reveals a slight difference. The odds of being vaccinated for males compared to females are 0.854, indicating slightly lower odds for males. However, this difference is not statistically significant. Within the cohort, the odds of being vaccinated for males are 0.908, suggesting a slightly lower likelihood compared to the overall population. On the other hand, the odds of being vaccinated for females in the cohort are 1.063, indicating a slightly higher likelihood compared to the overall population.

Lastly, exploring vaccination status and environment of origin, a notable difference emerges. The odds ratio for being vaccinated between individuals from urban and rural environments of origin is 0.371, implying significantly lower odds of being vaccinated for those from rural environments. For the cohort from urban environments, the odds of being vaccinated are 0.853, indicating a lower likelihood compared to the overall population. However, for the cohort from rural environments, the odds of being vaccinated increase significantly to 2.296, indicating a higher likelihood compared to the overall population.

Regarding sex and vaccination status, the odds ratio for being vaccinated between males and females is 0.895, suggesting slightly lower odds of vaccination for males. However, this difference is not statistically significant. Among males in the cohort, the odds of being vaccinated are 0.939, indicating a slightly lower likelihood compared to the overall population. Similarly, among females in the cohort, the odds of being vaccinated are 1.049, suggesting a slightly higher likelihood compared to the overall population.

Shifting focus to living area and vaccination status, a notable difference emerges. The odds ratio for being vaccinated between individuals living in urban and rural areas is 0.291, indicating significantly lower odds of being vaccinated for those residing in rural areas. For the cohort from urban areas, the odds of being vaccinated are 0.761, suggesting a lower likelihood compared to the overall population. However, for the cohort from rural areas, the odds of being vaccinated increase significantly to 2.615, indicating a higher likelihood compared to the overall population.

Examining the unfavorable evolution (death or transfer to intensive care) and vaccination status, the odds ratio for unvaccinated individuals is 0.805, suggesting slightly lower odds compared to the vaccinated group. However, the confidence interval is wide, indicating uncertainty. Among those with no unfavorable evolution in the cohort, the odds are 0.991, indicating a similar risk compared to the overall population. Conversely, for those with unfavorable evolution in the cohort, the odds increase to 1.231, suggesting a slightly higher risk, but with wide confidence intervals.

Analyzing antiviral treatment and vaccination status, the odds ratios for unvaccinated individuals receiving Remdesivir or Favipiravir/Molnupiravir indicate no significant difference in vaccination odds. The confidence intervals include 1, suggesting similar likelihoods compared to the vaccinated group.

Regarding immunosuppressive treatment and vaccination status, the odds ratio for unvaccinated individuals is 1.212, indicating no significant difference in vaccination odds. Among those not receiving immunosuppressive treatment in the cohort, the odds are 1.072, suggesting a similar likelihood compared to the overall population. For those receiving immunosuppressive treatment in the cohort, the odds are 0.885, indicating slightly lower odds of vaccination, but the confidence interval includes 1, indicating no significant difference.

Finally, when examining imaging changes and vaccination status, the odds ratios for unvaccinated individuals with alveolar pulmonary opacity or accentuated pulmonary interstitial changes indicate no significant difference in vaccination odds. The confidence intervals include 1, suggesting similar likelihoods compared to the vaccinated group.

In summary, the risks identified in unvaccinated individuals reveal some variations based on sex, living area, unfavorable evolution, antiviral treatment, immunosuppressive treatment, and imaging changes. Although slight differences exist in the vaccination odds for certain groups, statistical significance is not consistently observed. The wide confidence intervals in some cases suggest uncertainty and the need for further research.

Discuss major bias in paper.

4 - The authors mention justly that the calculation is incomplete due to inclusion bias at the emergency department (line 364-367). Assuming that people will present at the emergency department with serious disease, this could be a major inclusion bias, if vaccination also in Romania protects against significant disease. This point is the major weakness of this study and deserve to be discussed with literature data on vaccination protecting from disease at a more prominent place in the discussion, as it is central to this study.

The emergency department directed to the emergency hospital the cases where Covid-19 was not the main pathology, i.e. they had other medical or surgical emergencies, such as acute abdomen, trauma, vascular accident. I included this pathology in the exclusion criteria, and clarified it in the text of the discussions.

Round digits to significant numbers.
5a-In writing p-values 1-2 significant numbers suffices. Thus 0.04 and not 0.0440, 0.2 or 0.22 and not 0.2173.

5b- Table 3 should have less significant digits (e.g. max 3 for < 1000 patients), thus age 61.8 and not 61.798). Presentation of min, max & median is not essential.

Thanks for the suggestion. I rounded all the numbers in the text.

 Unclarity.

6- What is meant with "We must mention that not all patients had the complete set of data available. 111" Does this impact the results.

          We evaluated the missing data and they do not exceed 2% for each parameter. There are omissions of some analyzes that were collected recently in another clinic and that we did not take into account for the initial assessment of the patient upon admission.

Reviewer 4 Report

This is a work confirming previous observations, but it has a regional novelty aspect with well-done and described analysis.

1. As the Omicron variant prevailed during the study period, please include this in the title.

2. For the chapter "Materials and methods", please provide graphics with the flowchart and key data, e.g. subgroups with the number of patients, type of examinations, activities, etc. thanks to this, the work will be clear.

Line 68 - "Thus, we started from the hypothesis of a milder clinical evolution of COVID-19 cases […]" Better "milder clinical course" I guess. Same on line 71.

Line 78-79 - please provide the names and manufacturers of the tests used during the study period. If there are such data, the number qualified as positive by PCR and antigen test.

Line 83 – “evolution”? maybe “development”

Line 92 - "criteria: oxygen saturation <90% on room air [...]" - mental shortcut not understandable to everyone. Please clarify. Line 95-99. Only one patient was classified into groups? But is it plural?

Line 95 - shouldn't there be a colon? "Regarding the SARS-CoV-2 vaccination status, we considered the vaccinated patients: […]". "fully vaccinated patients: [...]" etc. Please correct the syntax to understandable in lines 95-100 so that there are no doubts in the division into individual groups.

Line 102 - information already partly given earlier in lines 78-79. Please do not repeat the same information. There should be criteria for inclusion in the study, earlier information should concern the tests used. It would be good to provide information on how many patients had only positive test results without other parameters or symptoms confirming the infection.

Line 113 - "unfavorable evolution"?

Line 117 - "Mean ± D.S (standard de-117 viation)" ?? or "Mean ± SD"

Line 120 - it's hard to remember what each group meant. It would be good to make a clear grouping in advance with a clear specification so that it can be easily and quickly found.

Line 132 - we use the median rather than the average for age. Both are measures of central values, but they indicate something different.

Table 1 - "Not vaccinated" or Unvaccinated?; "Mines; Max” or “Min-Max” ? A student's test was used - did the samples show a normal distribution?

 Minor editing of English language required

Author Response

Answers for REVIEWER 4

First of all, we would like to thank you very much for your review and consideration for our work. Thank you!

Here are our answers to the subjects you highlighted:

This is a work confirming previous observations, but it has a regional novelty aspect with well-done and described analysis.

  1. As the Omicron variant prevailed during the study period, please include this in the title.

Thanks for the suggestion, I added it like this:

Comparative evaluation of the clinical severity of COVID-19 of the vaccinated and unvaccinated patients, in Southeastern Romania, in the first 6 months of 2022, in the Omicron wave

  1. For the chapter "Materials and methods", please provide graphics with the flowchart and key data, e.g. subgroups with the number of patients, type of examinations, activities, etc. thanks to this, the work will be clear.

We added the flow chart and details in the text, in red.

Line 68 - "Thus, we started from the hypothesis of a milder clinical evolution of COVID-19 cases […]" Better "milder clinical course" I guess. Same on line 71.

Thanks for the observation, I changed in both places.

Line 78-79 - please provide the names and manufacturers of the tests used during the study period. If there are such data, the number qualified as positive by PCR and antigen test.

Yes, we added in text, in red.

Line 83 – “evolution”? may be “development”

Yes, I changed in red.

Line 92 - "criteria: oxygen saturation <90% on room air [...]" - mental shortcut not understandable to everyone. Please clarify. Line 95-99. Only one patient was classified into groups? But is it plural?

Thanks for the observation, I changed in red.

Line 95 - shouldn't there be a colon? "Regarding the SARS-CoV-2 vaccination status, we considered the vaccinated patients: […]". "fully vaccinated patients: [...]" etc. Please correct the syntax to understandable in lines 95-100 so that there are no doubts in the division into individual groups.

 Yes. I added a colon.

Line 102 - information already partly given earlier in lines 78-79. Please do not repeat the same information. There should be criteria for inclusion in the study, earlier information should concern the tests used. It would be good to provide information on how many patients had only positive test results without other parameters or symptoms confirming the infection.

We restructured the inclusion criteria for clarification.

Line 113 - "unfavorable evolution"?

Unfavourable course- unfavorable evolution -considered transfer to the intensive care unit (ICU) or death during hospitalization.

Line 117 - "Mean ± D.S (standard deviation)" ?? or "Mean ± SD"

Yes, I correct.

Line 120 - it's hard to remember what each group meant. It would be good to make a clear grouping in advance with a clear specification so that it can be easily and quickly found.

Yes, I did a flow chart.

Line 132 - we use the median rather than the average for age. Both are measures of central values, but they indicate something different.

We added the medians, is relevant.

Table 1 - "Not vaccinated" or Unvaccinated?; "Mines; Max” or “Min-Max” ? A student's test was used - did the samples show a normal distribution?

Yes , I correct. Yes the samples had a normal distribution.

Round 2

Reviewer 1 Report

 quaries has been rectified

Author Response

Answers for REVIEWER 1

First of all, we would like to thank you very much for your review and consideration for our work. Thank you!

  1.  Quaries has been rectified.

Reponse:

Thank you for the constructive comments and for the contribution to the improvement of this article.

Reviewer 2 Report

The Authors have revised the manuscript as per the suggestions.

Authors have included some discussion part of results ( pertaining to Odds ratio ) in result section. It may be better to  be included in the discussion section.

Addition  of suitable references will increase the strength of discussion. 

Minor modifications required

Author Response

Answers for REVIEWER 2

First of all, we would like to thank you very much for your review and consideration for our work. Thank you!

Here are our answers to the subjects you highlighted:

  1. The Authors have revised the manuscript as per the suggestions.
  2. Authors have included some discussion part of results (pertaining to Odds ratio) in result section. It may be better to be included in the discussion section.
  3. Addition  of suitable references will increase the strength of discussion. 

Response

Thank you for the recommendation. I took part of the interpretation of the results and added to the discussions, as well as some references, in green.

Reviewer 3 Report

Current interpretation seems to be based on the size of the risk estimate, but risk estimates should be interpreted using 95% confidence intervals e.g., if the whole interval is below 1.000 it is reduced, if all is above 1.000 it is increased, if it is spanning 1.000 (one value below, one above) no statistical certainty can given.

Could be clearer but English is okay.

Author Response

Answers for REVIEWER 3

First of all, we would like to thank you very much for your review and consideration for our work. Thank you!

  1. Current interpretation seems to be based on the size of the risk estimate, but risk estimates should be interpreted using 95% confidence intervals e.g., if the whole interval is below 1.000 it is reduced, if all is above 1.000 it is increased, if it is spanning 1.000 (one value below, one above) no statistical certainty can given.

Response

Thank you for the recommendation. I reformulated the results and discussions according to the confidence interval, in green.